# Impact of high temperatures on enzyme-linked immunoassay (ELISA) performance for leptin measurements in human milk stored under varied freeze/thaw conditions

Victoria Bertacchi[1]*, Margaret Corley[1,2], Gary P. Aronsen[1], Richard G. Bribiescas[1]

**1** Reproductive Ecology Laboratory, Department of Anthropology, Yale University, New Haven, Connecticut, United States of America, **2** Department of Ecology and Evolutionary Biology, Yale University, New Haven, Connecticut, United States of America

* victoria.harries@yale.edu

## Abstract

Ambient temperature conditions are a common concern during laboratory analysis. Due to unexpected shipping conditions, leptin ELISA kits (Leptin Ultrasensitive, ALPCO USA; Catalog #22-LEPHUU-E01) arrived from the manufacturer at our laboratory at a temperature (76.3°F/24.6°C) well above the 2-8°C conditions recommended by the manufacturer. Since no data are available on the effects of high ambient temperature exposure on the performance of this commercial assay, we opportunistically assessed assay performance using human milk samples. Leptin measurement of recently collected and frozen human milk samples was compared between the warm temperature exposed assay kits and Normal kits that arrived and were stored at recommended temperatures (2-8 °C). We found that assay kit exposure to warm temperature during shipping resulted in sample results that were significantly different from Normal kits despite similar standard curve performance. Measurement variability from human milk samples increased with warmed kits in association with greater freeze/thaw times. This suggests that even under high temperature transportation conditions, this leptin assay performance is robust with kit reagents but compromised with human milk samples. We conclude that kits exposed to high temperature during shipment and/or storage should not be used to run human milk samples and that our concerns may extend to other biological media (i.e., serum, urine, or saliva). This study fills a critical gap in the literature on assay performance validation under non-ideal conditions, such as high temperatures. As global temperatures continue to rise, this question will become more pertinent to research integrity if left unaddressed. In light of our findings, we propose that industry standards for ELISA kit shipping and handling should be evaluated to ensure that all kits are being received in an optimal condition.

## Introduction

Commercial shipping of items requiring cool-/cold-storage presents an array of challenges [1,2]. While, ambient temperature conditions are a common concern during laboratory

**Data availability statement:** The data underlying the results presented in the study are available at https://www.openicpsr.org/openicpsr/project/211841/version/V2/view

**Funding:** The author(s) received no specific funding for this work.

**Competing interests:** The authors have declared that no competing interests exist.

analysis, proper shipping and storage conditions are required to ensure that hormone assay plates and reagents remain stable and valid. Enzyme-linked Immunoassay (ELISA) kits contain an array of temperature-sensitive reagents allowing for the testing of hormone concentrations and endocrine function [3,4]. As ELISAs are relatively easy to perform, produced in bulk to provide for both consistency and accuracy in testing, and formulated to work on multiple biological media (i.e., serum, saliva, urine), they are very commonly used in laboratory settings [5]. However, these require effective and efficient cool-/cold-storage shipping logistics. An example of one such ELISA test is leptin in human milk samples.

Leptin is a metabolic hormone that is predominately synthesized in white adipose tissue and found ubiquitously throughout the body [6–8]. The hormone plays a key role in energy balance and appetite regulation, with implications for health risks such as obesity, metabolic syndrome, and type 2 diabetes [8–10]. Due to the increased burden of obesity and metabolic diseases across the world, leptin has received a surge of research across many disciplines – including medicine, public health, and anthropology. In particular, interest has increased into exploring the presence of leptin in human milk given that breastfeeding actively transfers the hormone concentration from mother to infant [11–13]. This transfer of leptin has implications for infant growth and development, energy homeostasis, and early nutritional programming [13–19]. As a result, leptin in human milk has emerged as an important biomarker of lactation and breastfeeding studies [20].

Due to the combination of an unexpected heatwave and a delayed shipment journey, our research lab was presented with the opportunity to test the impact of exposure to high ambient temperature on leptin ELISA plate readings. A shipment of two leptin ELISA kits (Leptin Ultrasensitive, Catalog #22-LEPHUU-E01, ALPCO USA, Salem, NH) was received by co-author GPA at our New Haven, CT laboratory via a commercial shipper. The shipment occurred during a heatwave affecting the northeastern United States of America. As is protocol in our laboratory for receiving temperature sensitive material, the package was opened upon arrival and inspected using a temperature probe (Oregon Scientific Digital Thero-Clock; Model no.: NAW881EXT) to record the internal temperature. Multiple measurements reached an internal temperature of 76.3 °F (~24.6 °C), well above the recommended manufacturer storage temperature of 2-8 °C.

While information is present regarding the maintenance of ELISA temperatures within the recommended range during the actual sample processing [5,21], no information is available – at least publicly – detailing what repercussions will occur if the kits are exposed to temperature during storage prior to analysis. This led us to query whether or not transportation and/ or storage conditions (especially those outside the range recommended by the manufacture) impacts the quality and consistency of sample analysis and should therefore be a reason for concern. After communication with ALPCO, they provided replacement kits of the same lot number which arrived cold to the touch and with frozen gel-packs. This created the conditions to opportunistically test the impact of high temperature exposure on assay performance on human milk measurements utilizing an existing study examining changes in leptin measurements over a thawing period. A paper by Harries et al. [22] provides more information on the background to the original research design and impetus for measuring leptin in human milk samples.

This study is the first to our knowledge to test the impact of high temperature exposure on leptin ELISA kits using human milk samples. Our aim for this study is to provide information for researchers that will allow them to make informed decisions regarding the handling of ELISA kits that have been exposed to high temperatures. We hypothesize that the temperature conditions would make no difference to the sample readings between the two plates (acceptable temperature range versus high temperature exposure).

## Methods

Approval for the study was granted by Yale University's Institutional Review Board #2000027635. Written, informed consent was gained from participants. Data collection took place July 2020.

An ongoing experiment gave us the opportunity to compare results from the high temperature exposed assay kits with kits that had remained within the temperature range recommended by the manufacturer. Measurements of leptin concentration in samples exposed to various hours of thawing, a common field and transportation challenge, were conducted to determine the consistency of leptin values [22]. Human milk samples (n = 56; 4 participants with 14 samples each) recently collected for the study were assayed using the high temperature exposed ("Warm") and recommended temperature range ("Normal") assay kits.

Full methods for sample collection, experiment description, and sample processing are reported in Harries et al. [22]. In brief, four participants gave a 100ml sample of human milk which was separated into 20 identical aliquots of 5ml. These aliquots were stored at -80°C. The experiment pertinent to this paper involved removing 13 samples per participant from the freezer and placing them in a cooler. One sample per participant remained in the freezer conditions to be used as a baseline. Starting at the four-hour mark, one sample per participant was returned to the freezer conditions. Four hours is the typical time that human milk takes to defrost in a cooler with ice packs, hence the decision to begin the return to the freezer at this hour [23]. See Fig 1 for study design. After all samples had returned to a frozen condition, they were thawed, processed, and assayed to analyze hormonal degradation rate across the thawing period.

Samples were assayed using a commercially available leptin ELISA kit (Leptin Ultrasensitive, Catalog #22-LEPHUU-E01, ALPCO USA, Salem, NH). This leptin assay kit is marketed for use with human milk samples, with a sensitivity of 0.01 ng/ml. Samples were run in duplicate using the Warm and Normal kits on the same day by the same researcher (VB). Two plates were run for each condition, Participants 1 and 2 were run on one plate (referred to as Normal Plate 1 and Warm Plate 1) and Participants 3 and 4 were run on the second (referred to as Normal Plate 2 and Warm Plate 2). All standards, quality controls, buffers, and reagents

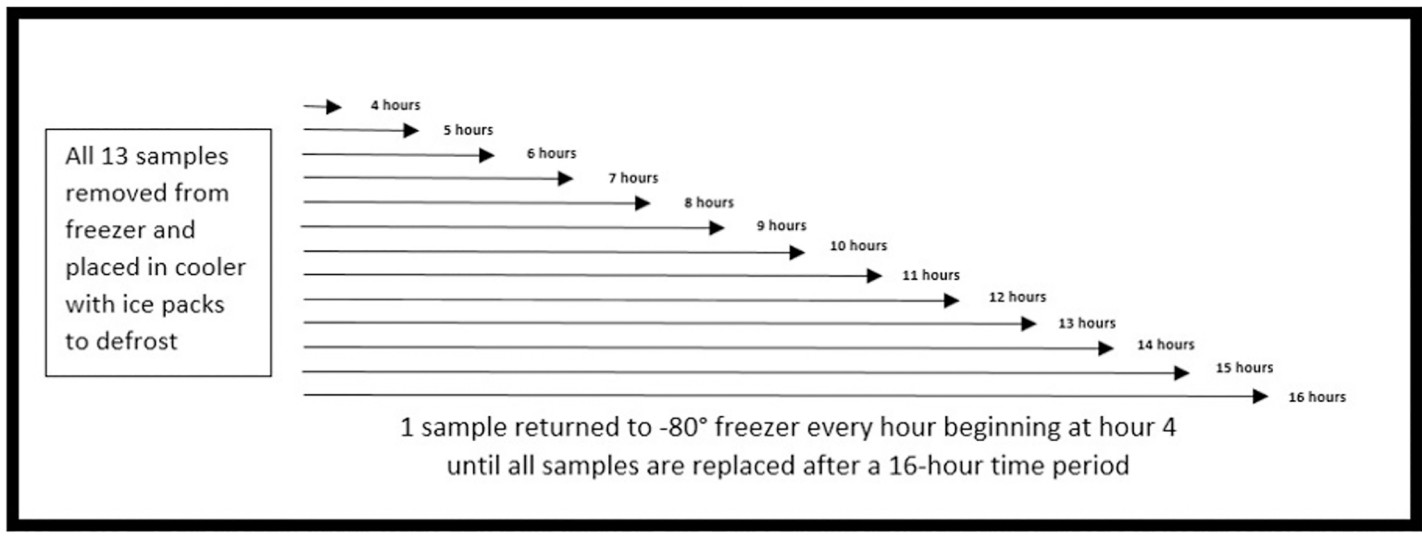

**Fig.1. Study design. Adapted from Harries et al. 2024.**

were used only for their corresponding kits. Kit lot numbers were the same for all four assay kits (lot #030521). All protocols were performed following the ALPCO instructions provided with the kits. The plates were read using the Byonoy 'Absorbance 96' microplate reader, using their data reduction software (software version 1.1.0). Inter-assay coefficient of variance (CV%) was calculated using the high- and low-quality controls (QC) provided in the assay kits. An inter-assay QC variation below 15% is typically deemed acceptable, while the acceptable value for intra-assay CV% is below 10%. Inter-assay QC variation between the plates was 9.7% (Normal) and 30.6% (Warm). The mean intra-assay CV% was 8.7% for Normal Plate 1, 9.4% for Warm 1, 9.6% for Normal 2, and 14.4% for Warm 2.

Sample measurements were obtained by averaging duplicate sample wells. Statistical analysis was completed in R 4.0.4, using packages ggplot2 and ggpubr [24]. An ANOVA was run to test variance of standard means between the Normal and Warm plates. Percentage change from baseline concentration to hour-16 concentration was calculated for each participant for comparison between plates. Percentage difference across the entire sample was calculated using the highest and lowest readings across the thaw period for Normal vs. Warm plate samples per participant. Leptin levels for each participant were plotted to compare concentration readings between plates across the study thawing period. Bland-Altman plots were graphed to assess agreement between the samples, comparing those analyzed on the Normal vs. Warm plates. The plots include three reference lines; the overall mean percentage difference (highest and lowest readings) between the sample readings on each plate and the upper and lower limits of agreement (calculated as mean + or – 1.96 SD, respectively).When using Bland-Altman plots, a result of 0 indicates no change between the samples [25,26].

## Results

In kits from both conditions (Warm/Normal), blank wells read below detectability and all standards and QCs read within manufacturer recommendations. Assay performance comparisons are presented (Table 1).

However, sample measurements differed notably between the two plate conditions (Table 2).

**Table 1. Concentration of standards (ng/ml).**

| Sample | Normal Plate 1 Reading | Normal Plate 2 Reading | Warm Plate 1 Reading | Warm Plate 2 Reading | Mean (SD) | |
|---|---|---|---|---|---|---|
| Blank (Absorbance) | 0.010 | 0.010 | 0.011 | 0.010 | 0.010 (0.001) | |
| STD 1 | 0.058 | 0.045 | 0.044 | 0.056 | 0.051 (0.01) | $F = 0.00$; |
| STD 2 | 0.478 | 0.512 | 0.509 | 0.483 | 0.495 (0.02) | $p = 1.00$ |
| STD 3 | 1.528 | 1.488 | 1.492 | 1.520 | 1.507 (0.02) | |
| STD 4 | 3.472 | 3.508 | 3.507 | 3.483 | 3.492 (0.02) | |
| STD 5 | 5.015 | 4.997 | 4.996 | 5.008 | 5.00 (0.01) | |

**Table 2. Mean concentration of hormone values for Normal and Warm Plates.**

| Participant | Normal Plate Leptin Mean (SD); ng/ml | Warm Plate Leptin Mean (SD); ng/ml |
|---|---|---|
| 1 | 0.17 (0.01) | 0.35 (0.16) |
| 2 | 0.69 (0.05) | 0.85 (0.25) |
| 3 | 0.13 (0.02) | 0.33 (0.23) |
| 4 | 0.50 (0.06) | 0.36 (0.19) |
| Mean | 0.37 (0.02) | 0.42 (0.12) |

Sample thawing duration resulted in marked measurement variability in the Warm plates compared to Normal plates (Fig 2).

The percentage change between the baseline and hour 16 samples were calculated to evaluate hormone degradation across the study period. Results are presented in Table 3.

The percentage difference between all samples across the study period per participant were calculated to evaluate differences in hormone calculation for Normal vs. Warm plates. Results are presented in Table 4.

To evaluate the agreement between the Normal and Warm plate samples, a Bland-Altman plot was created (Fig 3). To calculate the percentage difference between the samples, the difference for the participant on the Normal vs. the Warm plate (presented in Table 4) were compared.

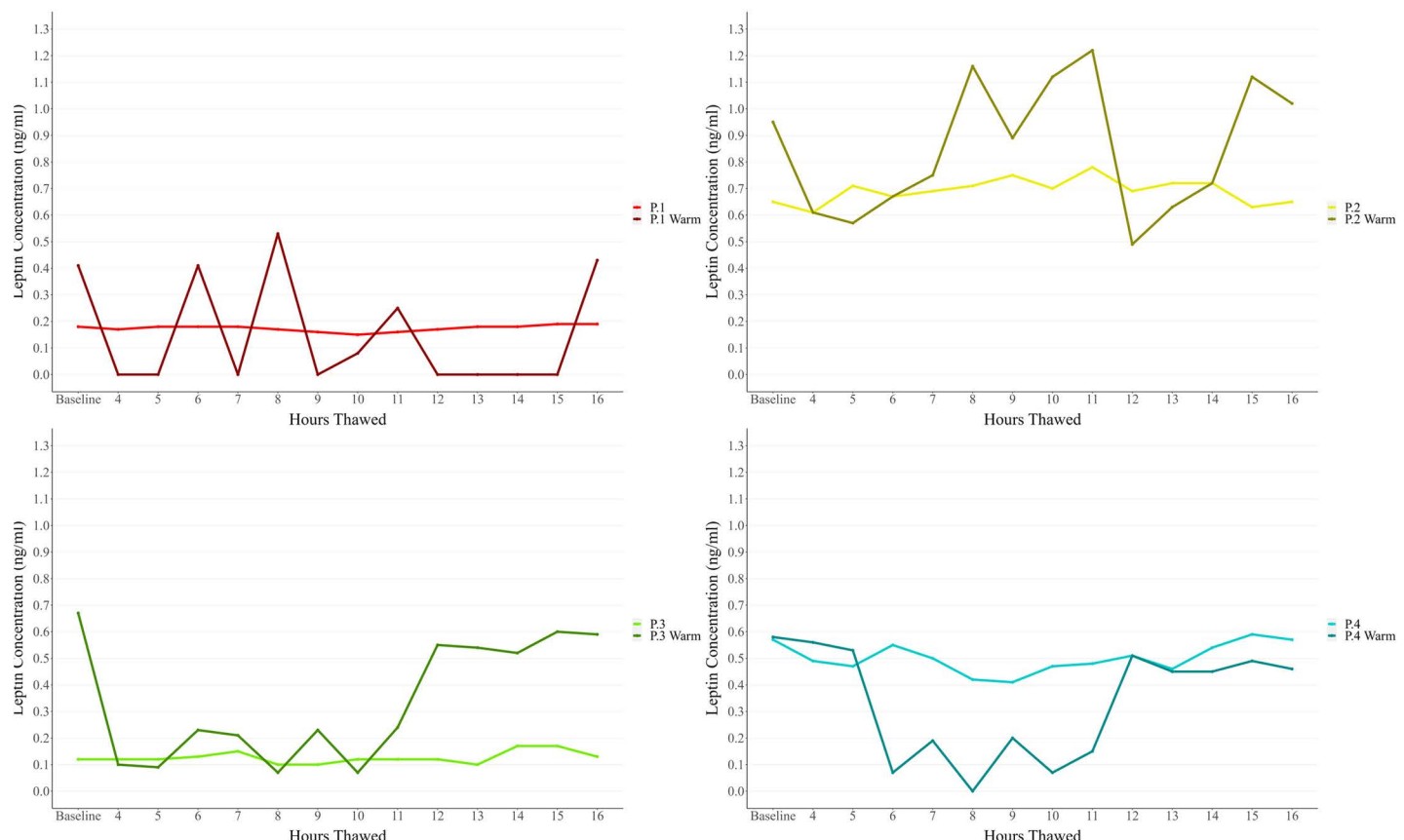

**Fig 2. Comparison of leptin concentrations (ng/ml) across hours thawed between Normal and Warm temperature ELISA plate outcomes.**

Table 3. Percentage hormonal change over sample thaw period.

| Participant | Normal Plate | Warm Plate |
|---|---|---|
| 1 | +3.3% | +5.9% |
| 2 | 0.0% | +6.5% |
| 3 | +5.0% | -11.4% |
| 4 | +0.4% | -19.3% |
| Mean | +2.2% | -4.6% |

**Table 4. Percentage difference in hormone concentration reading over sample thaw period.**

| Participant | Normal Plate | Warm Plate |
|---|---|---|
| 1 | 21.1% | 84.9% |
| 2 | 21.8% | 59.8% |
| 3 | 35.3% | 89.6% |
| 4 | 28.8% | 87.9% |
| Mean | 26.7% | 80.6% |

In a Bland-Altman plot, a mean of 0% would indicate no difference between the sample readings on the Normal versus the Warm plate. From the Confidence Intervals (CI) we can see that 95% of the percentage differences between the Normal and Warm plate samples are expected to fall in the range of -31.79% to -75.80% with a mean of -53.8% difference between the same samples depending on the plate used.

## Discussion

The results do not support our hypothesis that there would be no difference in the sample readings between the two plates (high temperature exposed versus acceptable temperature range). Instead, our results indicate that exposure to higher temperatures during transportation beyond manufacturer recommendations results in notably different leptin measurements in human milk samples as well as marked variability in concentration values despite overall consistency of standard curves, manufacturer QCs, and blank detectability in Normal versus Warm plates. This is interesting to note that if testing an ELISA for suspicion of high temperature conditions, the readings for the standards may still remain consistent, indicating contrasting assay reagent and/or antibody performance with biological media, in this case human milk. As QCs are often used as an indication of analysis quality, this potentially poses a serious issue. In this particular kit (Leptin Ultrasensitive, Catalog #22-LEPHUU-E01, ALPCO USA, Salem, NH), the provided standards are a recombinant human leptin and the QCs are human serum. Both are lyophilized. It may be that the recombination or lyophilization process provides a robustness to these reagents that enables them to be protected from high temperature exposure. Additionally, the use of lyophilized serum for the QCs may not be representative of reagent and/or antibody performance compared to the human milk samples. This highlights the need for awareness of the potential impact of high temperature exposure on ELISAs using an array of different media to evaluate for any differing plate stability during analysis.

Comparing the intra- and inter-assay coefficient of variance between the Normal and Warm plates shows higher discrepancies in the Warm plates than their counterparts. Both Warm plates consistently produced higher variation among duplicates, larger CV% values, and greater measurement variability (Fig 2) for all four subjects. All plates were run at the same time using the same lab conditions and researcher (VB), reducing observer error and/or other external causes. The observed pattern of higher CV% for Warm plates may be due to the impact of the high temperature lowering the precision and accuracy of the plates and impacting the reading of each well. The CV% difference is smaller when examining intra-assay compared to inter-assay variation – 8.7% for Normal Plate 1, 9.4% for Warm Plate 1, 9.6% for Normal Plate 2, and 14.4% for Warm Plate 2. These results suggest that the higher temperature affected the plate as a whole, as opposed to individual wells or areas. Furthermore, the inter-assay CV% values – 9.7% for the Normal plates and 30.6% for the Warm plates – indicate that the impact of the high temperatures makes inter-plate comparison challenging, regardless of consistent kit lot numbers.

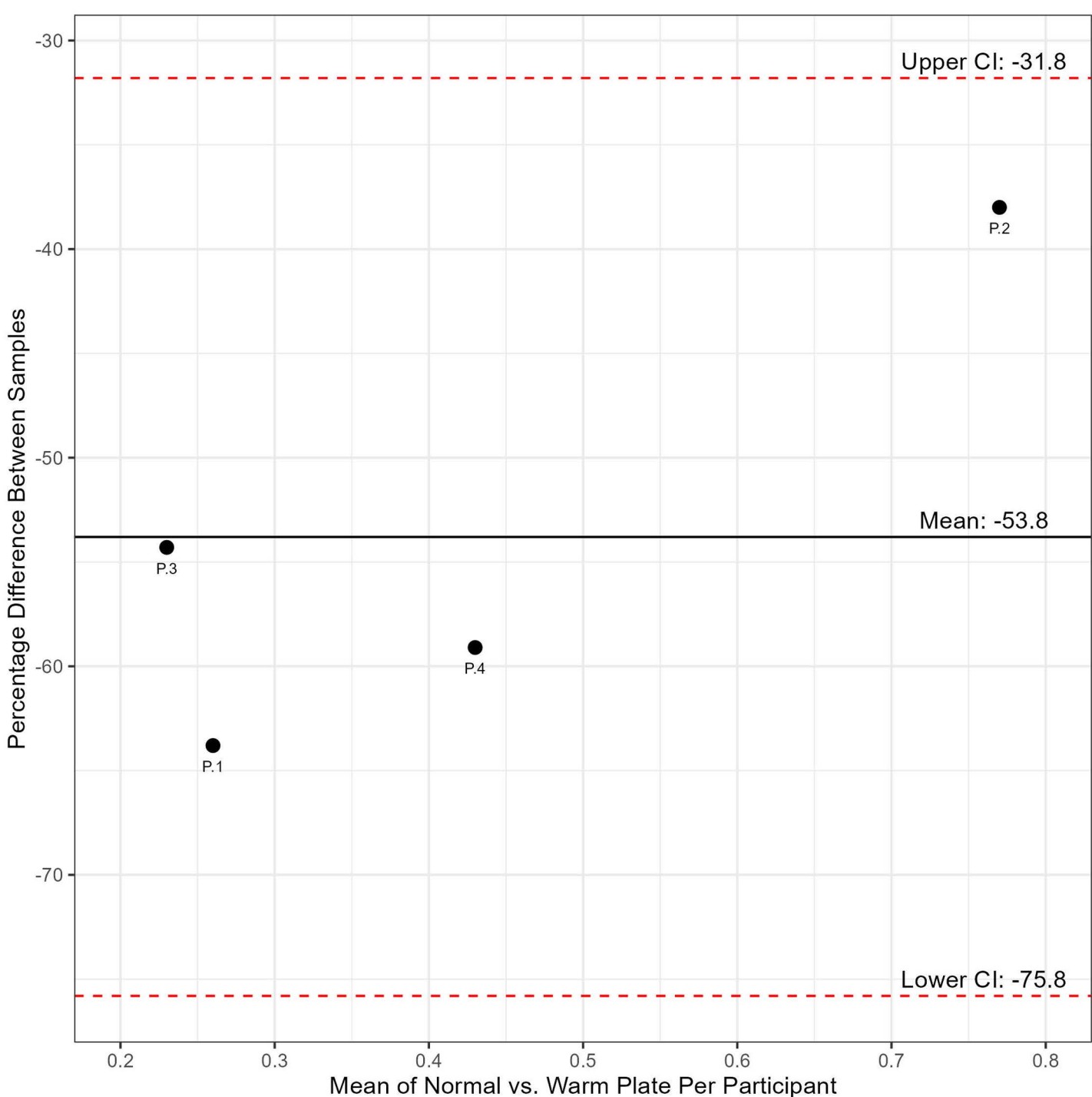

**Fig 3. Bland-Altman plot for percentage difference between normal and warm plate samples.**

Leptin concentration levels for each individual showed wide variation between the Warm and Normal plates. Normal plates showed a far more consistent pattern over time thawed, while the Warm plates showed marked spikes and troughs in contrast (Fig 2). The impact of heat also led to lower precision in leptin concentration measures in the samples, this resulted

in a mean SD of 0.12 (range: 0.16 to 0.25) in the Warm plate sample readings compared to a mean SD of 0.02 (range: 0.01 to 0.06) in the Normal plate (Table 2). Additionally, many readings from the Warm plate were below the detection level – despite a low analytical sensitivity yield of 0.01 ng/ml. This was particularly prominent in the samples for Participant 1 – who had the lowest mean concentration of leptin present in their human milk samples, therefore negating the utility of plates that have undergone high temperature exposure.

The percentage change over the sample thaw period (shown in Table 3) highlights the inconsistency of results when comparing between the Normal and Warm plates. The range of percentage change from baseline to Hour-16 for the Normal plates is narrower and more consistent across the four participants with little change over the study period (range: 0.0% to + 5.0%, mean: + 2.2%) compared to the Warm plates (range: -19.3% to + 6.5%, mean: -4.6%). Similarly, the percentage difference across the entire study period between the Normal vs. Warm plates that is shown in Table 4 and the Bland-Altman plot (Fig 3) highlights this discrepancy between the plate readings. In the Bland-Altman plot, the mean percentage difference between the samples was calculated as -53.8% – much lower than the ideal of 0%. When examining the confidence interval expected range (-31.79% to -75.80%), it is evident that the sample analysis outcomes cannot be consistently compared and experienced a high discrepancy in readings between the two plates despite being aliquoted from the same sample. The percentage differences seen across the different analyses – percentage change (Table 3), percentage difference (Table 4), and Bland-Altman plot (Fig 3) – are evidence that exposure to high temperatures did impact assay reliability. The result of this would mean concentration readings that are much higher or lower than those that should have been calculated in the sample that is not easily predicted or corrected for (as evidence by Fig 2). In practice, the consequences of this could range from reporting a much higher level of human variation in samples than what is actually present to wrongly diagnosing an individual.

The reasons as to why the Warm plates displayed higher concentration readings for the same samples are unclear and beyond the scope of this research design. However, we can hypothesize two mechanisms through which this variation occurred: 1) increased non-specific binding and 2) degradation of the antibody and/or conjugate reagent that becomes evident in human milk media. In terms of increased non-specific binding, we know that binding capacity increases with exposure to temperature [27–30]. As the temperature is raised, the increase in kinetic energy would create a greater capacity for non-specific binding on the assay plate resulting in a higher concentration reading from the same sample. Additionally, it is likely that the heat did not impact the binding capacity on the plate ubiquitously explaining why we see differences across a plate reading for the same sample – despite the concentration being identical in principle.

Alternatively, all of the reagents included with the assay kit are temperature sensitive, therefore, it is possible that the high temperature exposure caused proteolytic degradation in one – if not all – of these components. This degradation process could impact the reagents either through a breakdown or a restructuring of the chemical compounds that subsequently alter the activity of the antibody [31–34]. When subjected to heat, the chemical bonds that hold together the structure of the molecules can weaken and break due to the increased energy, resulting in instability of the remaining fragments thus making them altered and/or unstable for use in analysis [35–39]. As the particular assay used for this experiment is a "sandwich" ELISA, the kits contain two antibodies; a mouse anti-leptin antibody that pre-coats the plate and a conjugate that is added after the sample incubation period which is comprised of a mix of biotinylated mouse-anti-human leptin antibody and horseradish peroxidase conjugated streptavidin. We do not know how the antibodies bind as this is proprietary information. However, it could be theorized that either as the plate was exposed to high

temperatures the affinity of the antibody that coats the plate was increased or that degradation of the conjugate resulted in less competition for the sample. Either of these scenarios would result in a higher variation of concentration reading that we see present in the Warm plates compared to the Normal counterparts. It would be of interest in the future to isolate and test each element of the ELISA kit individually in order to identify which are being impacted by the high temperature exposure.

As climate change will continue altering ambient temperatures locally and globally, manufacturers, shippers, and laboratories will need to take additional steps to insure effective and consistent temperature controls for their products. Additional steps may include measures such as more sophisticated temperature monitors, expedited shipping methods, and/or more effective quality control measures; examples of these can be seen already in industries such as food and agriculture [1,2]. Importantly, with rising and increasingly unpredictable temperatures becoming a potential burden on sample transportation and storage, it presents a new and increasing expense for all involved in research and may impact researchers with limited funds – such as students or those in early career years.

There are several strengths to this research. The study design – which consisted of several aliquots of only four samples – lends itself well to the comparison between the plates, allowing for easier comparison of discrepancies in the Warm plate readings. Additionally, that all plates were run side-by-side by the same researcher (VB) helps to lower any bias or inconsistencies that may have occurred if samples were run in differing environments or with different laboratory techniques or equipment. However, the opportunistic nature presents some limitations as well. The small sample size and plate number means that our results may not be generalizable. Testing the influence of temperature exposure on ELISA reliability in a larger number of participants would enable us to ensure that the results we have found in this experiment are not due to chance. Additionally, only one hormone (leptin) was tested in our research, utilizing only one sample media (human milk) for which leptin concentration is particularly low. As the hormone concentration is typically low, it may have exacerbated the disruption of the concentration readings by the ELISA plate as a result of the high temperature exposure. It would be of interest to undertake the same experiment investigating the influence of temperature exposure on sample readings using hormones and/or media for which the concentration is much higher and therefore subject to less competition with the antibodies of the assay. In particular, hormones that are some of the most commonly studied due to their implications for health and reproductive function – for example: cortisol, insulin, thyroid-stimulating hormone (TSH), melatonin, testosterone, estrogen, and prolactin – would be a priority due to the implications of falsely measuring these hormones for medical treatment and outcomes. Additionally, the research of metabolic hormones in human milk is a growing and exciting field, therefore we would suggest that testing the robustness of assay plates subjected to temperature fluctuations for the measurement of adiponectin, ghrelin, and insulin in human milk would be of great interest to the field of lactation and breastfeeding. We feel that this research serves to opens up conversation for awareness of the limitations of hormone analysis and presents an opportunity for future research to further explore the impact of hot temperatures on ELISA outcomes. Of interest would be a larger sample size, an array of different hormones, more mediums of samples, and/or differing assay plates from additional manufacturers.

Despite the limitations, we believe that the results of this study are important for the scientific community and present many actionable suggestions for assay companies and researchers. For the assay companies and manufacturers, our findings suggest that care must be taken in guarantying that their product delivery line takes place under optimal conditions alongside plans that ensure temperature control and regulations are in place. This would help make sure that deliveries of assay plates arrive at the laboratories in a consistent condition. For

the researchers and laboratories, our findings suggest that there is a need for close attention to be paid to temperature conditions of the kits that are produced externally and shipped to the facilities in order to ensure that reliable data is obtained. In addition, we emphasize that researchers should not rely on assay curve performance to assess potential heat damage, as we found that reagent and sample performance are not always aligned – potentially due to the different media of all reagents. Instead, our recommendations for researchers are to ensure vigilance in recording temperatures of assay deliveries upon arrival, especially after experiencing shipping delays or holdups. As researchers become more reliant on the use of commercial kits and/or components, it is necessary to become more vigilant of the conditions in which they arrive. If a delivery is found to be above that which is deemed stable by the manufacturing company, our suggestion is to discard all impacted kits due to the risk of incorrect measurement of the hormone of interest.

In conclusion, the Warm plates showed more variation, different measures, and lower quality than the Normal plates, regardless of the standard curve and QC consistency. When this particular ELISA kit is exposed to temperatures well above manufacturer recommendations, 1) it should not be used to run human milk samples, and 2) standard curve and manufacturer QC performance data are insufficient for assessing human milk measurement consistency and accuracy. This study fills a critical gap in the literature on assay performance validations under non-ideal conditions such as high temperature, opening up discussions for other sensitive biological media. As global temperatures continue to rise, this question will become more pertinent to research integrity if left unaddressed. In light of our findings, we propose that industry standards for ELISA kit shipping and handling should be evaluated to ensure that all kits are being received in an optimal condition. This opportunistic investigation focuses on one particular commercial assay kit as well as one biological media (human milk). Additional investigations using other commercial and proprietary assay kits/protocols, hormones, and biological media (i.e., serum, urine, or saliva) are merited. While commercial assay kits provide many benefits, the proprietary nature of commercial assay performance data presents challenges. This study highlights the necessity for researchers to validate and monitor consistency between plates above that of the standard curve and, importantly, stresses the need for greater awareness of assay and sample conditions as a source of measurement variation.

## Acknowledgments

We would like to thank Ashley Minihan for her assistance in finding participants through her local breastfeeding support group.

## Author contributions

**Conceptualization:** Victoria Bertacchi, Gary P. Aronsen, Richard G. Bribiescas.

**Formal analysis:** Victoria Bertacchi.

**Investigation:** Victoria Bertacchi, Margaret Corley.

**Methodology:** Victoria Bertacchi.

**Resources:** Gary P. Aronsen, Richard G. Bribiescas.

**Supervision:** Richard G. Bribiescas.

**Visualization:** Victoria Bertacchi.

**Writing – original draft:** Victoria Bertacchi.

**Writing – review & editing:** Victoria Bertacchi, Margaret Corley, Gary P. Aronsen, Richard G. Bribiescas.

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
