## [Decision Letter · Decision Letter 0]

23 Sep 2024

PONE-D-24-18339Impact of high temperatures on enzyme-linked immunoassay (ELISA) performance for leptin measurements in human milk stored under varied freeze/thaw conditionsPLOS ONE

Dear Dr. Bertacchi,

Thank you for submitting your manuscript to PLOS ONE. After careful consideration, we feel that it has merit but does not fully meet PLOS ONE’s publication criteria as it currently stands. Therefore, we invite you to submit a revised version of the manuscript that addresses the points raised during the review process.

We look forward to receiving your revised manuscript.

Kind regards,

Tommaso Lomonaco, Ph.D

Academic Editor

PLOS ONE

Journal Requirements:

2. Please include a separate caption for each figure in your manuscript.

Reviewers' comments:

Reviewer's Responses to Questions

**Comments to the Author**

1. Is the manuscript technically sound, and do the data support the conclusions?

Reviewer #1: Partly

Reviewer #2: Partly

Reviewer #3: Yes

2. Has the statistical analysis been performed appropriately and rigorously? 

Reviewer #1: N/A

Reviewer #2: I Don't Know

Reviewer #3: Yes

3. Have the authors made all data underlying the findings in their manuscript fully available?

Reviewer #1: Yes

Reviewer #2: Yes

Reviewer #3: Yes

4. Is the manuscript presented in an intelligible fashion and written in standard English?

Reviewer #1: Yes

Reviewer #2: No

Reviewer #3: Yes

5. Review Comments to the Author

Reviewer #1: The manuscript titled “Impact of high temperatures on enzyme-linked immunoassay (ELISA) performance for leptin measurements in human milk stored under varied freeze/thaw conditions” is interesting and useful both for kit companies (so that their deliveries to laboratories take place under optimum conditions) and for operators and researchers (so that they pay attention to the temperature conditions of the kits they use in order to obtain reliable data).

I have my doubts that it can be included as a research article in PLosOne Journal: it is a sort of validation of a protocol referring to room temperature, which does not provide any additional information of data; moreover, despite the interesting study and the usefulness it has for the scientific community, the very few references reported almost suggest that it is a report, or protocol not an article. Check whether it is possible to convert to other forms of publication permitted by the PLosOne Journal.

Given these premises, there are the following major imperfections:

Lines 65: change the sentence: “For more information on background to the study and the impetus for measuring leptin in human milk samples, see (7).” The reference 7 cannot be indicated that way in the text. For example, authors could write: “A study by Harries et al. in 2024, provides more information on the background to the research and the impetus for measuring leptin in human milk samples (7).”

Line 75: “milk samples (n = 56; 4 participants with 14 samples each)”. The 14 samples of the 4 subjects were obtained at different times or at different stages? please specify.

Lene 77: “Methods for sample collection and sample processing can be found here: (7)”. Rewrite this sentence reporting briefly the method used ….as reported by…..

Lines 84-87: “Quality control (QC) variation between the plates was 9.7% (Normal) and 30.6% (Warm). The mean intra-assay coefficient of variance (CV%) was 8.7% for Normal Plate 1, 9.4% for Warm 1, 9.6% for Normal 2, and 14.4% for Warm 2.” What are these values? Are they reported by manufacture? In the materials and methods section, it is necessary to report what was measured and how (if possible, with the formulas used) therefore:

1) Quality control (QC) variation between the plates

2) Sample thawing measurement variability in the Warm plates compared to Normal plates.

3) The percentage change between the baseline

4) The percentage difference between all samples across the study period

5) the percentage differences between the Normal and Warm plate samples

6) intra- and inter-assay coefficient of variance between the Normal and Warm plates.

All of these aspects (which are some of the parameters used to validate a method) are reported in the results, and are discussing in the discussion section, but not listed in the materials and methods, and no explanation of how they are calculated is given.

Furthermore, despite the study being a comparison of the same samples assayed with kits stored at different temperatures, how come it did not go into detail with other parameters normally used for the validation of an ELISA method (Andreasson U, Perret-Liaudet A, van Waalwijk van Doorn LJC, Blennow K, Chiasserini D, Engelborghs S, et al. A practical guide to immunoassay method validation. Front Neurol. (2015) 6:179. doi: 10.3389/fneur.2015.00179)? e.g. parallelism, dilution, reproducibility, accuracy, precision. Even if the kit is sold for the milk matrix and was therefore not indispensable analysing these parameters, they would have strengthened the results.

Reviewer #2: General comment

This study provides interesting data on the effect of high temperature on the performance of a commercial ELISA test of leptin measurements in human milk. However, neither the presented information, nor the applied experiment or the obtained data of this study can provide solid conclusion for the author hypothesis. In addition, this study has many fundamental limitations that rendered the acceptance in the current form is very difficult.

Abstract and introduction

- The presented data and information in both section is very poor and more relevant information is needed including the extent of use of the tested ELISA, leptin as marker for milk characteristics testing, previous similar reports of effect of temperature changes or other environmental or storage factors on some biological markers.

- Also, novelty and aspect of application of obtained knowledge should be clearly described in the abstract.

- The aim of this study is ambiguous and not clear.

Materials and methods

- The research design is not clear and the authors should support their concept and used criteria for evaluation using evidence of previous reports or validation trials.

- Using samples from 4 participants only is very risky in the evaluation and interpretation of the obtained data.

Results and discussion

- The explanation of results section is very poor and scanty information was added.

- The conflict in the results of obtaining consistent data for standards and inconsistent data for test samples combined with using samples from 4 participants only confirmed the doubts on the obtaining the same conclusion by other researchers.

- What is the author explanation for obtaining higher leptin concentration in warm plates than those normal plates in Table 2?

Reviewer #3: Manuscript titled "Impact of high temperatures on enzyme-linked immunoassay (ELISA) performance for leptin measurements in human milk stored under varied freeze/thaw conditions."

The study addresses the effects of high ambient temperatures on the performance of leptin ELISA kits used to analyze human milk samples. The manuscript provides a valuable and timely assessment of the impact of high temperatures on ELISA performance in human milk samples since temperature effects during shipping are a common issue in laboratory settings but are often overlooked. The focus on human milk samples adds further relevance to maternal and child health research. The study is well-designed, but expanding the sample size, discussing the underlying biochemical mechanisms in more detail, and providing practical guidelines for researchers would strengthen the paper. My main concerns are follows:

1) The novelty could be better highlighted by positioning the study as filling a critical gap in assay performance validation under non-ideal conditions. Emphasizing the potential impact on other sensitive biological media (e.g., serum, urine) would broaden the study's significance.

2) The experimental design is clear, with appropriate use of control (Normal) and high-temperature-exposed (Warm) ELISA kits. The use of duplicate measurements and side-by-side comparisons is a strength of the study. However, the small sample size (four participants) may limit the generalizability of the results. A larger sample size and inclusion of additional biological media would strengthen the findings and make the results more applicable across different research contexts. Additionally, more information on the exact freeze/thaw conditions for each sample would enhance reproducibility.

3) The data presentation is generally clear, with tables and figures that effectively compare leptin measurements between Warm and Normal plates. The use of Bland-Altman plots to assess agreement between plates is appropriate and well-explained. Some figures, particularly the Bland-Altman plot, could benefit from clearer labeling and a more detailed explanation of how to interpret the reference lines. The discussion of percentage differences between Warm and Normal plates could be expanded to include more context on how these differences impact assay reliability in practice.

4) The study finds that high temperatures during shipment significantly affected the precision and accuracy of leptin measurements in human milk samples, with Warm plates exhibiting greater variability. These findings are important but require a deeper exploration of the underlying mechanisms. For instance, how do high temperatures affect the stability of reagents in ELISA kits, and why is the impact more pronounced in human milk samples compared to controls? The discussion could delve deeper into the biochemical reasons for the increased variability in Warm plates. Drawing on literature related to enzyme stability under heat stress or reagent degradation would provide a more comprehensive understanding of the results.

5) The manuscript discusses the potential implications of high-temperature exposure on ELISA performance in human milk samples, extending these concerns to other biological media. This is an important point, but the discussion would benefit from more concrete recommendations for researchers and laboratories. The authors should consider providing specific guidelines for researchers, such as when to reject ELISA kits exposed to high temperatures, how to verify kit integrity post-shipping, or alternative solutions for mitigating temperature-related issues. This would give the study more practical value.

6) The authors acknowledge the limitations of the study, particularly the small sample size and focus on a single hormone (leptin) in one biological medium (human milk). They also propose future research directions, including testing additional hormones and biological media. The limitations could be expanded to discuss the variability in ELISA kit performance across different manufacturers or assay types. A more detailed outline of future research priorities, including large-scale studies and different sample types, would strengthen this section.

7) The conclusions are aligned with the results and provide a clear take-home message: ELISA kits exposed to high temperatures should not be used for human milk samples, and researchers should not rely solely on standard curve and QC performance to assess kit validity under these conditions. Strengthen the conclusion by discussing how these findings could influence industry standards for ELISA kit shipping and handling, particularly in the context of rising global temperatures.

6. PLOS authors have the option to publish the peer review history of their article (what does this mean? ). If published, this will include your full peer review and any attached files.

**Do you want your identity to be public for this peer review?** For information about this choice, including consent withdrawal, please see our Privacy Policy .

Reviewer #1: No

Reviewer #2: No

Reviewer #3: No

---

## [Author Response · Author response to Decision Letter 1]

25 Nov 2024

We would like to thank the three reviewers for their comments. We believe that the edits that have been made to the manuscript greatly improve the content and quality of this proposed article. Below we have included replies to each reviewer’s suggestions and comments. Our responses are in italics.

Reviewer #1: The manuscript titled “Impact of high temperatures on enzyme-linked immunoassay (ELISA) performance for leptin measurements in human milk stored under varied freeze/thaw conditions” is interesting and useful both for kit companies (so that their deliveries to laboratories take place under optimum conditions) and for operators and researchers (so that they pay attention to the temperature conditions of the kits they use in order to obtain reliable data).

I have my doubts that it can be included as a research article in PLosOne Journal: it is a sort of validation of a protocol referring to room temperature, which does not provide any additional information of data; moreover, despite the interesting study and the usefulness it has for the scientific community, the very few references reported almost suggest that it is a report, or protocol not an article. Check whether it is possible to convert to other forms of publication permitted by the PLosOne Journal.

Thank you for your comment. We will add in additional language to ensure that the manuscript better meets the criteria for a research article.

Given these premises, there are the following major imperfections:

Lines 65: change the sentence: “For more information on background to the study and the impetus for measuring leptin in human milk samples, see (7).” The reference 7 cannot be indicated that way in the text. For example, authors could write: “A study by Harries et al. in 2024, provides more information on the background to the research and the impetus for measuring leptin in human milk samples (7).”

Thank you, this change has been made.

Line 75: “milk samples (n = 56; 4 participants with 14 samples each)”. The 14 samples of the 4 subjects were obtained at different times or at different stages? please specify.

We have included details for this both in the line you have indicated and, in the addition, we made to the methods.

Lene 77: “Methods for sample collection and sample processing can be found here: (7)”. Rewrite this sentence reporting briefly the method used ….as reported by…..

Thank you for this suggestion, we have included more details on methods.

Lines 84-87: “Quality control (QC) variation between the plates was 9.7% (Normal) and 30.6% (Warm). The mean intra-assay coefficient of variance (CV%) was 8.7% for Normal Plate 1, 9.4% for Warm 1, 9.6% for Normal 2, and 14.4% for Warm 2.” What are these values? Are they reported by manufacture? In the materials and methods section, it is necessary to report what was measured and how (if possible, with the formulas used) therefore:

1) Quality control (QC) variation between the plates

2) Sample thawing measurement variability in the Warm plates compared to Normal plates.

3) The percentage change between the baseline

4) The percentage difference between all samples across the study period

5) the percentage differences between the Normal and Warm plate samples

6) intra- and inter-assay coefficient of variance between the Normal and Warm plates.

All of these aspects (which are some of the parameters used to validate a method) are reported in the results, and are discussing in the discussion section, but not listed in the materials and methods, and no explanation of how they are calculated is given.

We apologize for their absence from the methods section. This information was included in the original paper that is referenced (Harries et al. 2024). We have included the information in this paper as well to help the reader. Inter-assay quality control and intra-assay CV% are reported in the methods section alongside the updated information on the assay plate that was used for measurement. We have not reported the sample thawing variability, the percentage change, or the percentage differences in the methods as these were the variables of interest and therefore are reported in the results and discussed later in the discussion. The explanation of how percentage change and percentage difference were calculated are included in the last paragraph of the methods section.

Furthermore, despite the study being a comparison of the same samples assayed with kits stored at different temperatures, how come it did not go into detail with other parameters normally used for the validation of an ELISA method (Andreasson U, Perret-Liaudet A, van Waalwijk van Doorn LJC, Blennow K, Chiasserini D, Engelborghs S, et al. A practical guide to immunoassay method validation. Front Neurol. (2015) 6:179. doi: 10.3389/fneur.2015.00179)? e.g. parallelism, dilution, reproducibility, accuracy, precision. Even if the kit is sold for the milk matrix and was therefore not indispensable analysing these parameters, they would have strengthened the results.

We have added in the information on the assay kits that were used during this analysis. The company lists breastmilk as a suitable media for this particular assay. While we acknowledge that it is important for in-house validation of immunoassay methods, it was unfortunately beyond the scope of this study. Therefore, the deployment of this particular assay for use with breastmilk was provided by the assay company. In the protocol handbook for the assay is listed: sensitivity, specificity, reproducibility and precision, linearity, recovery, and interference. For the purpose of this experiment, we deemed these an acceptable level of validation for the ELISA kit use.

Reviewer #2: General comment

This study provides interesting data on the effect of high temperature on the performance of a commercial ELISA test of leptin measurements in human milk. However, neither the presented information, nor the applied experiment or the obtained data of this study can provide solid conclusion for the author hypothesis. In addition, this study has many fundamental limitations that rendered the acceptance in the current form is very difficult.

Abstract and introduction

- The presented data and information in both section is very poor and more relevant information is needed including the extent of use of the tested ELISA, leptin as marker for milk characteristics testing, previous similar reports of effect of temperature changes or other environmental or storage factors on some biological markers.

This was the impetus for this paper, as there is not a lot of data on the effect of temperature changes/environmental or storage factors on ELISA assay outcomes for concentration readings of biological markers. This is what led us to believe that querying whether or not transportation conditions (especially outside the range recommended by manufacture) impacts the quality of sample analysis was an important study. We have added more information regarding this and regarding leptin as a marker in milk into the introduction to highlight this.

- Also, novelty and aspect of application of obtained knowledge should be clearly described in the abstract.

We believe that the novelty and aspect of application are clearly stated in the abstract (eg. lines 23-24: “Since no data are available on the effects of high ambient temperature exposure on the performance of this commercial assay…” and lines 32-34 “We conclude that kits exposed to high temperature during shipment and/or storage should not be used to run human milk samples and that our concerns may extend to other biological media (ie., serum, urine, or saliva)”).

- The aim of this study is ambiguous and not clear.

We have edited the introduction (and discussion) to include the hypothesis for the study. Thank you for drawing attention to the ambiguity that was created through its absence.

Materials and methods

- The research design is not clear and the authors should support their concept and used criteria for evaluation using evidence of previous reports or validation trials.

We apologize for the absence of more detail regarding the research design. The design was fully detailed in the original study paper that was referenced within the text. We realize now that this was not sufficient. More details and a figure have been added into the paper.

- Using samples from 4 participants only is very risky in the evaluation and interpretation of the obtained data.

We completely understand that four participants are not a large sample size and acknowledge this within the paper. As this was an opportunistic study that happened alongside another experiment, we did not have control over the sample size. We have added in additional disclaimer within the paper that the sample size is very small and that more studies with larger number of participants are needed to explore this further.

Results and discussion

- The explanation of results section is very poor and scanty information was added.

- The conflict in the results of obtaining consistent data for standards and inconsistent data for test samples combined with using samples from 4 participants only confirmed the doubts on the obtaining the same conclusion by other researchers.

- What is the author explanation for obtaining higher leptin concentration in warm plates than those normal plates in Table 2?

We have taken your comments into consideration and have included more information in the discussion section on why we think we obtained higher leptin concentration in the warm plates compared to the normal plates

Reviewer #3: Manuscript titled "Impact of high temperatures on enzyme-linked immunoassay (ELISA) performance for leptin measurements in human milk stored under varied freeze/thaw conditions."

The study addresses the effects of high ambient temperatures on the performance of leptin ELISA kits used to analyze human milk samples. The manuscript provides a valuable and timely assessment of the impact of high temperatures on ELISA performance in human milk samples since temperature effects during shipping are a common issue in laboratory settings but are often overlooked. The focus on human milk samples adds further relevance to maternal and child health research. The study is well-designed, but expanding the sample size, discussing the underlying biochemical mechanisms in more detail, and providing practical guidelines for researchers would strengthen the paper. My main concerns are follows:

1) The novelty could be better highlighted by positioning the study as filling a critical gap in assay performance validation under non-ideal conditions. Emphasizing the potential impact on other sensitive biological media (e.g., serum, urine) would broaden the study's significance.

We have added additional language regarding the novelty of the study.

2) The experimental design is clear, with appropriate use of control (Normal) and high-temperature-exposed (Warm) ELISA kits. The use of duplicate measurements and side-by-side comparisons is a strength of the study. However, the small sample size (four participants) may limit the generalizability of the results. A larger sample size and inclusion of additional biological media would strengthen the findings and make the results more applicable across different research contexts. Additionally, more information on the exact freeze/thaw conditions for each sample would enhance reproducibility.

Thank you for this comment. We completely understand that four participants are not a large sample size and acknowledge this within the paper. As this was an opportunistic study that happened alongside another experiment, we did not have control over the sample size nor the ability to expand to a larger sample size later. We have added in additional disclaimer within the paper that the sample size is very small and that more studies with larger number of participants are needed to explore this further. Additionally, while the sample information is included in the original paper that we reference towards for more information, we realize that this exclusion was a mistake. More details on sample experiment conditions and a figure detailing it have been added into this paper.

3) The data presentation is generally clear, with tables and figures that effectively compare leptin measurements between Warm and Normal plates. The use of Bland-Altman plots to assess agreement between plates is appropriate and well-explained. Some figures, particularly the Bland-Altman plot, could benefit from clearer labeling and a more detailed explanation of how to interpret the reference lines. The discussion of percentage differences between Warm and Normal plates could be expanded to include more context on how these differences impact assay reliability in practice.

We have added more language into the Bland-Altman results and the discussion on percentage differences as per your suggestion.

4) The study finds that high temperatures during shipment significantly affected the precision and accuracy of leptin measurements in human milk samples, with Warm plates exhibiting greater variability. These findings are important but require a deeper exploration of the underlying mechanisms. For instance, how do high temperatures affect the stability of reagents in ELISA kits, and why is the impact more pronounced in human milk samples compared to controls? The discussion could delve deeper into the biochemical reasons for the increased variability in Warm plates. Drawing on literature related to enzyme stability under heat stress or reagent degradation would provide a more comprehensive understanding of the results.

Thank you, we have added this into the discussion.

5) The manuscript discusses the potential implications of high-temperature exposure on ELISA performance in human milk samples, extending these concerns to other biological media. This is an important point, but the discussion would benefit from more concrete recommendations for researchers and laboratories. The authors should consider providing specific guidelines for researchers, such as when to reject ELISA kits exposed to high temperatures, how to verify kit integrity post-shipping, or alternative solutions for mitigating temperature-related issues. This would give the study more practical value.

Thank you, we have included some more actionable suggestions in our discussion.

6) The authors acknowledge the limitations of the study, particularly the small sample size and focus on a single hormone (leptin) in one biological medium (human milk). They also propose future research directions, including testing additional hormones and biological media. The limitations could be expanded to discuss the variability in ELISA kit performance across different manufacturers or assay types. A more detailed outline of future research priorities, including large-scale studies and different sample types, would strengthen this section.

We have added in additional ideas that include outlines for future research priorities per your suggestion.

7) The conclusions are aligned with the results and provide a clear take-home message: ELISA kits exposed to high temperatures should not be used for human milk samples, and researchers should not rely solely on standard curve and QC performance to assess kit validity under these conditions. Strengthen the conclusion by discussing how these findings could influence industry standards for ELISA kit shipping and handling, particularly in the context of rising global temperatures.

Thank you, we have added this into the conclusion following your suggestion.

---

## [Decision Letter · Decision Letter 1]

18 Feb 2025

Impact of high temperatures on enzyme-linked immunoassay (ELISA) performance for leptin measurements in human milk stored under varied freeze/thaw conditions

PONE-D-24-18339R1

Dear Dr. Bertacchi,

We’re pleased to inform you that your manuscript has been judged scientifically suitable for publication and will be formally accepted for publication once it meets all outstanding technical requirements.

Kind regards,

Tommaso Lomonaco, Ph.D

Academic Editor

PLOS ONE

Reviewers' comments:

Reviewer's Responses to Questions

**Comments to the Author**

1. If the authors have adequately addressed your comments raised in a previous round of review and you feel that this manuscript is now acceptable for publication, you may indicate that here to bypass the “Comments to the Author” section, enter your conflict of interest statement in the “Confidential to Editor” section, and submit your "Accept" recommendation.

Reviewer #1: All comments have been addressed

2. Is the manuscript technically sound, and do the data support the conclusions?

Reviewer #1: Yes

3. Has the statistical analysis been performed appropriately and rigorously? 

Reviewer #1: Yes

4. Have the authors made all data underlying the findings in their manuscript fully available?

Reviewer #1: Yes

5. Is the manuscript presented in an intelligible fashion and written in standard English?

Reviewer #1: Yes

6. Review Comments to the Author

Reviewer #1: Dear aurtori, I had asked you not to submit your work as a Research Article, but as another form such as a report or anything else planned by plosOne.

7. PLOS authors have the option to publish the peer review history of their article (what does this mean? ). If published, this will include your full peer review and any attached files.

**Do you want your identity to be public for this peer review?** For information about this choice, including consent withdrawal, please see our Privacy Policy .

Reviewer #1: No

---

## [Editor Report · Acceptance letter]

PONE-D-24-18339R1

PLOS ONE

Dear Dr. Bertacchi,

I'm pleased to inform you that your manuscript has been deemed suitable for publication in PLOS ONE. Congratulations! Your manuscript is now being handed over to our production team.

Kind regards,

on behalf of

Prof. Tommaso Lomonaco

Academic Editor

PLOS ONE